# Self-dual $S_3$-invariant quantum chains

**Edward O'Brien[1] and Paul Fendley[1,2]**

**1** Rudolf Peierls Centre for Theoretical Physics, Parks Rd, Oxford OX1 3PU, United Kingdom
**2** All Souls College, Oxford, OX1 4AL, United Kingdom

## Abstract

We investigate the self-dual three-state quantum chain with nearest-neighbor interactions and $S_3$, time-reversal, and parity symmetries. We find a rich phase diagram including gapped phases with order-disorder coexistence, integrable critical points with $U(1)$ symmetry, and ferromagnetic and antiferromagnetic critical regions described by three-state Potts and free-boson conformal field theories respectively. We also find an unusual critical phase which appears to be described by combining two conformal field theories with distinct "Fermi velocities". The order-disorder coexistence phase has an emergent fractional supersymmetry, and we find lattice analogs of its generators.

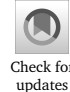

# 1 The model and the phase diagram

Since even before the Time of Landau, a common strategy in statistical mechanics has been to stipulate the symmetries of the system and construct the simplest model obeying them. With a $\mathbb{Z}_2$ symmetry, this approach yields the much-studied Ising model. The resulting critical point separating ordered and disordered phases in the two-dimensional classical case and the one-dimensional quantum chain is *self-dual* [1]. "Parafermionic" quantum chains with $n$ states per site and $\mathbb{Z}_n$ symmetry [2] are natural generalizations that have been intensively studied recently, for reasons including the appearance of topological order and potential experimental realizations [3].

However, very little systematic exploration of the simplest and most symmetric parafermionic chains has been done, a shame given their importance. We here aim to rectify the situation by analyzing a one-parameter family of three-state quantum chains with $S_3$ symmetry and Kramers-Wannier self-duality. These are the most general such chains with nearest-neighbor interactions invariant under time-reversal and spatial parity. Duality here is neither unitary nor invertible, as for example it maps the ordered ground states of the Potts chain to the unique ground state of the disordered phase. For the self-dual couplings we study, this non-triviality allows for novel phase transitions to occur [4]. Here we study the entire self-dual line and find a rich variety of previously unknown critical and gapped phases.

The Hamiltonian of our $L$-site chain written in terms of operators $\sigma_j$ and $\tau_j$ for $j = 1, \ldots, L$, all acting on the $3^L$ dimensional Hilbert space. Each operator acts non-trivially only on a single site $j$, e.g. $\sigma_2 = 1 \otimes \sigma \otimes 1 \cdots 1$, so that operators based on different sites commute. They obey

$$\sigma_j^2 = \sigma_j^\dagger, \qquad \tau_j^2 = \tau_j^\dagger, \qquad \sigma_j^3 = \tau_j^3 = 1, \qquad \sigma_j \tau_j = \omega \tau_j \sigma_j, \tag{1}$$

with $\omega = \exp(2\pi i/3)$. In a $\sigma_j$-diagonal basis,

$$\sigma = \begin{pmatrix} 1 & 0 & 0 \\ 0 & \omega & 0 \\ 0 & 0 & \omega^2 \end{pmatrix}, \qquad \tau = \begin{pmatrix} 0 & 0 & 1 \\ 1 & 0 & 0 \\ 0 & 1 & 0 \end{pmatrix}. \tag{2}$$

A convenient pair of single-site operators are the standard $su(2)$ generators for a spin-1 system:

$$S_j^+ = \frac{1}{3}\left(2 - \omega \tau_j - \omega^2 \tau_j^\dagger\right)\sigma_j^\dagger, \qquad S_j^- = S_j^{+\dagger}, \qquad S_j^z = \frac{i}{\sqrt{3}}\left(\tau_j^\dagger - \tau_j\right). \tag{3}$$

Key relations these obey are

$$(S_j^+)^3 = (S_j^-)^3 = 0, \qquad \left[S_j^z, S_j^\pm\right] = \pm S_j^\pm. \tag{4}$$

The Hamiltonian we study is self-dual under Kramers–Wannier duality, with duality-broken cases analyzed in [4]. The action of duality on the Hilbert space is given in a convenient and general form by using topological defects [5]. In the case of interest here, we simplify matters by studying only its action on translation-invariant Hamiltonians, where the operators transform as

$$\tau_j \to \sigma_j^\dagger \sigma_{j+1}, \qquad \sigma_j^\dagger \sigma_{j+1} \to \tau_{j+1}. \tag{5}$$

The self-dual quantum 3-state Potts Hamiltonian with periodic boundary conditions is the simplest one invariant under (5), namely

$$H_P = -\sum_{j=1}^{L}\left(\sigma_j^\dagger \sigma_{j+1} + \tau_j + \text{h.c}\right), \tag{6}$$

where $\sigma_{L+1} \equiv \sigma_1$. This Hamiltonian is the quantum spin-chain limit of the integrable self-dual 3-state Potts model [6,7]. Another nearest-neighbor self-dual Hamiltonian is [4]

$$H_1 = \sum_{j=1}^{L} \left( 3S_j^+ S_{j+1}^- - 3S_j^{+2} S_{j+1}^{-2} + \tau_j + \text{h.c.} \right). \tag{7}$$

In this form the self-duality is not obvious, but it is easily verified by rewriting the $S_j^\pm$ in terms of the $\tau_j$ and $\sigma_j$ using (3). In addition to being self-dual, both Hamiltonians are invariant under parity and time-reversal symmetries, namely $\mathcal{P} : \sigma_j \to \sigma_{L+1-j}$, $\tau_j \to \tau_{L+1-j}$, and $\mathcal{T} :$ $\sigma_j \to \sigma_j^\dagger$, $\tau_j \to \tau_j$. Under the latter, complex numbers are conjugated as well.

The Hamiltonian we study is an arbitrary linear combination of these two:

$$H(\theta) = \lambda_P H_P + \lambda_1 H_1 , \tag{8}$$

where a convenient coupling $\theta$ is defined by setting $\lambda_{\text{P}} \equiv \cos\theta$ and $\lambda_1 \equiv \sin\theta$. Writing $H(\theta)$ in terms of Temperley–Lieb generators (see e.g. [4]) gives the same expression as in Ref. [8] but in a different representation with different physics. Other similar Hamiltonians [9, 10] are distinct as well. In addition to $\mathcal{P}$ and $\mathcal{T}$, $H(\theta)$ is invariant under an $S_3$ permutation symmetry generated by charge conjugation and a $\mathbb{Z}_3$ cyclic shift symmetry. Charge conjugation acts on the operators by sending $\sigma_j \leftrightarrow \sigma_j^\dagger$, $\tau_j \leftrightarrow \tau_j^\dagger$, while the shift is generated by $\sigma_j \to \omega\sigma_j$, $\tau_j \to \tau_j$. Acting on the Hilbert space, shifts are generated by $\omega^Q$ with

$$Q = \sum_{j=1}^{L} S_j^z , \qquad \omega^Q = \prod_{j=1}^{L} \tau_j . \tag{9}$$

Manifestly, $\omega^Q$ commutes with the Hamiltonian, anticommutes with charge conjugation and obeys $(\omega^Q)^3 = 1$. The expression (8) gives the most general self-dual nearest-neighbor Hamiltonian invariant under all these symmetries.

The Hamiltonian $H_1$ by itself is a particular case of the integrable spin-1 XXZ chain [11]. Even beyond that, it has some very special properties. As is obvious from the form (7), it commutes with $Q$ from (9) itself, promoting the $\mathbb{Z}_3$ to a full $U(1)$ symmetry. Acting with duality (5) on $Q$ gives another $U(1)$ charge $\widehat{Q}$, which also must commute with the self-dual $H$. However, it is easy to check that $[Q, \widehat{Q}] \neq 0$ and that the two generate the non-Abelian Onsager algebra, resulting in large degeneracies [12]. Moreover, $H_1$ also has a "dynamical" lattice supersymmetry as it obeys $H_1 = \mathcal{Q}^2$, with a fermionic $\mathcal{Q}$ that changes the number of sites [13].

We do our analysis using detailed conformal field theory (CFT) and numerical techniques. Our results for the phase diagram of $H(\theta)$ are summarized in the phase diagram in Figure 1. All four individual Hamiltonians $\pm H_{\text{P}}$ and $\pm H_1$ are critical and integrable [11, 14], but their linear combination (8) is not integrable and not always critical. Four critical phases dominate the diagram, but very interesting gapped regions occur as well.

Three of the four large critical phases are described by well-known CFTs, as explained in section 2. Both the ferromagnet $H_P$ and and antiferromagnet $-H_P$ extend to critical phases. The former is described by the well-known $c = \frac{4}{5}$ three-state Potts CFT [15], a phase we dub "Potts 1". The latter is described by a $c = 1$ CFT, and below we explain why it describes a full region, not immediately obvious as the corresponding critical point in the classical square-lattice antiferromagnet is unstable [16]. Another region, the "Potts 2" phase, is also described by the $c = \frac{4}{5}$ CFT, and describes a transition between (duality-broken) phases with representation symmetry-protected topological (RSPT) order [4]. The integrable point with Hamiltonian $-H_1$ separates the Potts 1 and 2 phases along the self-dual line. This point is described by the

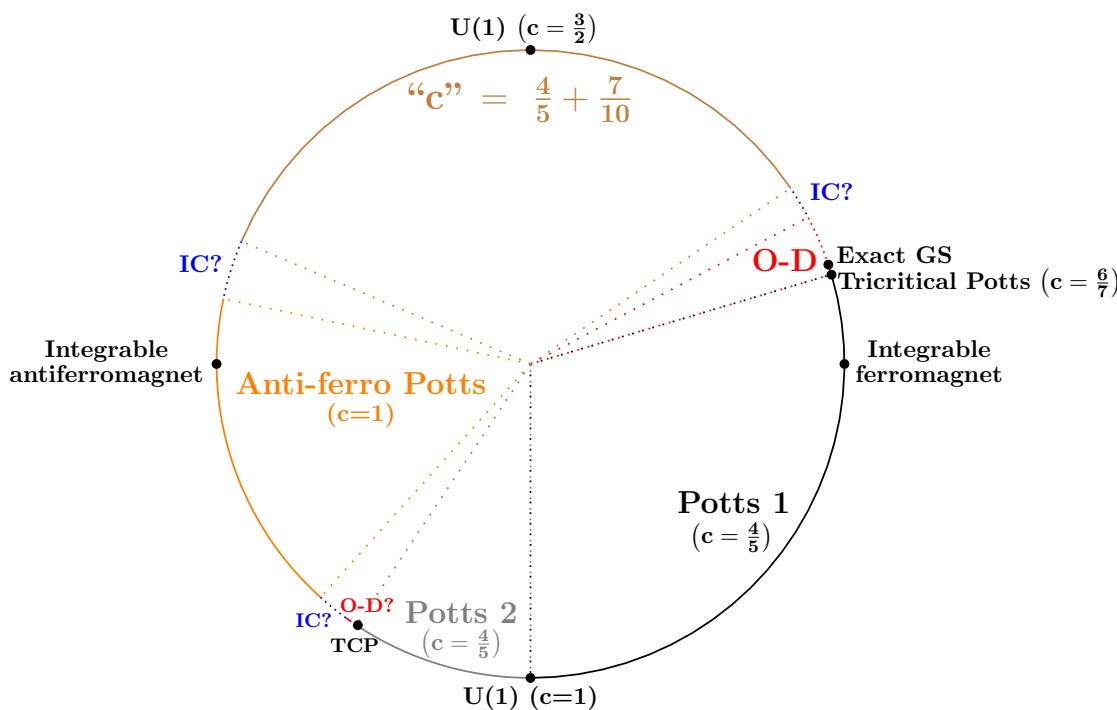

Figure 1: The phase diagram of Hamiltonian (8), including four large critical regions, one or two gapped phases with order-disorder (O-D) coexistence, and possible gapless incommensurate (IC) phases.

same $c = 1$ CFT as in the antiferromagnetic phase, but here perturbing it causes a flow to the Potts CFT [17, 18].

The fourth gapless phase, the "c"$=\frac{4}{5} + \frac{7}{10}$ phase described in section 3, is novel. The critical integrable point $H_1$ is described by a $c = \frac{3}{2}$ CFT [19, 20] as it is a special case of the integrable spin-1 chain [11]. This particular CFT can be decomposed into a product of two CFTs in a rather unusual way [21]. We show that while there are no relevant self-dual perturbations relevant under the symmetries, there exists a marginal one. While the spectrum remains gapless throughout a large region, the physics is not described simply by a CFT. We explain how it instead is best thought of as a combination of two interacting CFTs with different "Fermi velocities".

Another striking consequence of the self-duality is the existence of two gapped phases described in section 4. Both feature order-disorder coexistence, and occur after the Potts phases terminate in $c = \frac{6}{7}$ tricritical points. The phase beyond Potts 1 is governs a transition between conventional $\mathbb{Z}_3$ order and disorder, generalizing in a natural way the corresponding first-order phase transition in the $\mathbb{Z}_2$-invariant Majorana-Hubbard chain [22, 23]. The other gapped phase is even more uncommon, describing the coexistence between not-$A$ and RSPT order. Another property we explain is the presence of an unusual fractional supersymmetry.

## 2 The Potts phases

A key tool in our analysis is the knowledge of all scaling dimensions in the CFTs describing the continuum limit of the integrable points. In the region of such a critical point, the long-distance behavior is governed by an effective field theory found by perturbing the CFT by any relevant or marginal operators invariant under self-duality and all the lattice symmetries. When there are no such operators, the same CFT must describe an entire phase. The extent

of these regions can be determined in some cases by exploiting knowledge of flows between CFTs, while in others we must resort to numerical analysis. In this section we start by showing how three of the four critical phases can be obtained by such arguments.

## 2.1 The first Potts phase

The Hamiltonian $H(0) = H_P$ describes the self-dual ferromagnetic three-state Potts chain. It is integrable [14], and its continuum limit is described by a minimal CFT with $c = \frac{4}{5}$ [24]. No relevant self-dual operator obeying the symmetries of $H(\theta)$ exists in the Potts CFT, with the least irrelevant such operator having dimension $\frac{14}{5}$ [25]. Thus in the region of $H_P$, perturbing by $H_1$ must be irrelevant, and the Potts CFT continues to describe $H(\theta)$ for $|\theta|$ small. In figure 1, we dub this phase the "Potts 1" phase.

Effective field theories provide a nice way to understand the transitions out of this Potts phase. Namely, consider breaking the self-duality of $H_P$ by making the coefficients of the two types of terms in (6) unequal. When the coefficient of the first term is larger, $\langle \sigma_j \rangle \neq 0$ and the $S_3$ symmetry is spontaneously broken. The self-dual $H_P$ then describes a transition between order and disorder. Including the irrelevant perturbation $H_1$ does not change the situation, so this critical order-disorder phase transition persists along the self-dual Potts line. For positive $\theta$, perturbing by this irrelevant self-dual parity-invariant operator gives the same universality class as including vacancies in the $Q$-state Potts model, as discussed in depth for the $Q = 2$ Ising case in [22, 23]. For any $Q \leq 4$, one expects [26] that this phase terminates at tricritical point. The three-state tricritical Potts (TCP) model arising here is described by a CFT with $c = \frac{6}{7}$ [27]. In this CFT, there does exist a single relevant self-dual operator of dimension $\frac{10}{7}$ invariant under all symmetries of $H(\theta)$. The $c = \frac{6}{7}$ CFT thus can only describes a particular point in our phase diagram, and perturbing by this operator with the appropriate sign does indeed describe a flow from TCP to Potts [28, 29]. The Potts 1 phase therefore should terminate for $\theta$ positive at a TCP point.

We have confirmed this picture via DMRG [30, 31] using ITensor [32], locating the tricritical Potts point at $\lambda_1 \approx 0.297\lambda_P$, i.e. $\theta = \theta_{TCP} \approx 0.092\pi$. Our method is to measure the energies of low-lying levels, and exploit the fact that the energy levels in a CFT are directly related to the dimensions of the scaling operators creating them [33, 34]. Namely, the energy difference $E_a - E_b \propto (\Delta_a - \Delta_b)/L$, where $\Delta_a$ and $\Delta_b$ are the dimensions of operator creating the states labelled by $a$ and $b$ respectively. The ratio of any two energy differences

$$\frac{E_a - E_b}{E_c - E_d} = \frac{\Delta_a - \Delta_b}{\Delta_c - \Delta_d} , \tag{10}$$

is universal, and so we can compare the CFT results to our lattice simulations. We found that for $\theta \approx \theta_{TCP}$, they approach the TCP values as $L \to \infty$, while for $\theta$ smaller they approach the critical Potts values. In figure 2, we display one such ratio for $E_b = E_d = E_0^0$, $E_c = E_0^1$ and $E_a = E_1^1$, where $E_r^j$ its the energy of the $j$th excited state in the sector with $\mathbb{Z}_3$ charge $\omega^r$. The corresponding CFT scaling dimensions are respectively $0$, $\frac{4}{5}$, $\frac{17}{15}$ for Potts and $0$, $\frac{20}{21}$, $\frac{2}{7}$ for TCP, giving the ratios $\frac{17}{12}$ and $\frac{10}{3}$ respectively. Our DMRG computations of the scaling of the entanglement entropy [35] are also consistent with terminating the phase in a CFT with central charge $\frac{6}{7}$.

For the $\theta$ negative, a similar flow occurs. Here, however, we know the exact termination point, as both theoretical and numerical work shows that there are no phase transitions between the integrable points $H(-\frac{\pi}{2}) = -H_1$ and $H(0) = H_P$. The $U(1)$-invariant critical point $-H_1$ [11] terminating the phase is described by a free-boson CFT with $c = 1$ [36, 37]. Here a dimension $3/2$ operator obeys all the symmetries of $H(\theta)$, but not the $U(1)$ [4]. It is natural to identify this operator with $H_P$, and so the $U(1)$-invariant critical point is unstable. Perturbing

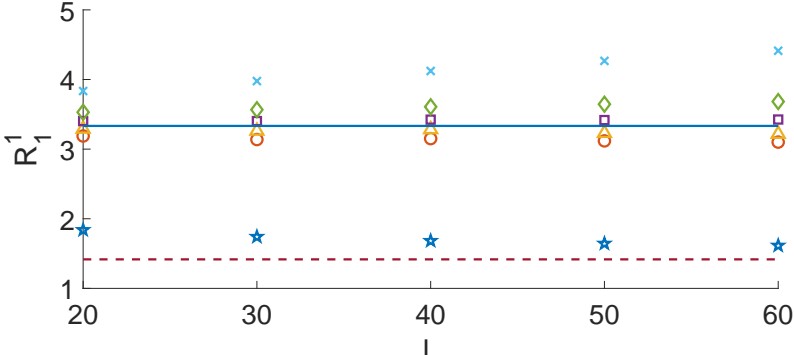

Figure 2: The ratio $R_1^1$ of energy differences (10) for $\lambda_P = 1$, with $E_b = E_d = E_0^0$, $E_c = E_0^1$ and $E_a = E_1^1$. The values of $\lambda_1$ are chosen near the TCP transition terminating the Potts 1 phase, with $\lambda_1 = 0.25$ (blue stars), 0.295 (red circles), 0.296 (yellow triangles), 0.297 (purple squares), 0.298 (green diamonds) and 0.3 (teal crosses). The dashed line is the critical Potts prediction of 17/12 while the solid line is the TCP prediction of 10/3.

by this operator in the field theory results in a flow to the $c = \frac{4}{5}$ fixed point [17, 18], with no intervening phases. Our numerical work confirms this picture in our lattice model, finding that for $-\frac{\pi}{2} < \theta < \theta_{\text{TCP}}$, all ratios from (10) approach the Potts CFT predictions as $L \to \infty$. The Potts 1 phase therefore extends to the entire lower-right portion of the phase diagram in figure 1.

## 2.2 The second Potts phase

An elegant bosonic field theory describes $H(\theta)$ in the region near $U(1)$-invariant critical point with Hamiltonian $-H_1$ [18, 38]. As detailed in [4], this effective field theory is the same for either sign of $\lambda_P$, implying that the same flow occurs on *both* sides of $\theta = -\frac{\pi}{2}$. Thus somewhat surprisingly, a second critical phase is described by same $c = \frac{4}{5}$ CFT for the ferromagnetic Potts critical point, even though $\lambda_P < 0$ means that the Potts Hamiltonian's contribution to $H(\theta < -\frac{\pi}{2})$ is antiferromagnetic. Breaking the self-duality shows that this Potts critical line describes an unusual transition, between "not-$A$" order, where two of the three directions of spin are favoured, and a representation symmetry-protected topological (RSPT) phase [4].

Even more remarkably, we find numerically that the second Potts phase terminates at the far end in the same way as the first phase. Increasing the magnitude of $\lambda_P$ while keeping it negative, we encounter another TCP point at $\lambda_P \approx 0.672\lambda_1 < 0$ ($\theta = \theta_{\text{TCP}'} \approx -0.69\pi$). One ratio (10) illustrating this behavior is shown in the figure 3. This phase and this termination occur just to the left of the $c = 1$ $U(1)$ point at the bottom of the phase diagram in figure 1.

## 2.3 Antiferromagnetic Potts phase

The third of the major critical regions surrounds the integrable antiferromagnetic three-state Potts (AFP) model $H(\pi) = -H_P$. At this integrable point, the long-distance description is a $c = 1$ free-boson CFT [16, 39] as at $\theta = -\frac{\pi}{2}$, although here the $U(1)$ symmetry is emergent. The self-duality and $S_3$ symmetry require that both have the same bosonic compactification radius [4].

An important distinction between the two integrable $c = 1$ points, however, is that the AFP Hamiltonian is stable under symmetry-preserving self-dual perturbations. The stability arises because the lattice analog of the relevant dimension-3/2 CFT operator has momentum

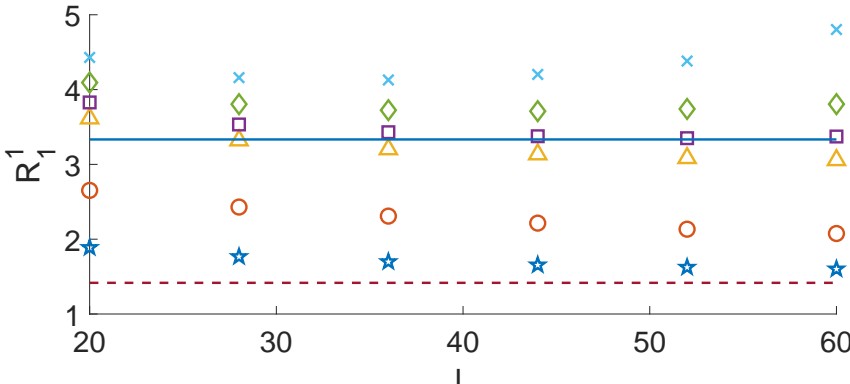

Figure 3: As in Figure 2, except for $\lambda_1 = -1$ to illustrate the termination of the second Potts phase. Points are at $\lambda_P = -0.6$ (blue stars), $-0.65$ (red circles), $-0.67$ (yellow triangles), $-0.672$ (purple squares), $-0.674$ (green diamonds) and $-0.676$ (teal crosses).

$\pi$ relative to the antiferromagnetic ground state, according to our numerics displayed in figure 4. As apparent, the ground state has momentum $k = \pi$ so that translation invariance is spontaneously broken, while the four states of dimension $3/2$ have lattice momentum $k = 0$. Since $H(\theta)$ is invariant under translation invariance, an operator with momentum $\pi$ relative to the ground state cannot appear in the effective field theory around $\theta = \pi$. All other relevant operators are disallowed as before, resulting in an AFP phase. This stability does not occur in the corresponding square-lattice classical model [16,39], presumably because its interactions are antiferromagnetic in both space and Euclidean time directions, while in our Hamiltonian setup, interactions in the "time" direction are effectively ferromagnetic. Our numerics indicate that the most likely scenario is that on both sides this antiferromagnetic phase terminates by an excited state crossing the ground state, resulting in gapless incommensurate phases. These crossings occur at around $\theta \approx 0.9\pi$ and at $\theta \approx -0.73\pi$, and the resulting small incommensurate regions are shown in figure 1 as "IC?".

# 3 The "c"$= \frac{4}{5} + \frac{7}{10}$ phase

The fourth large critical phase is quite unusual and interesting. We start by analyzing the integrable $U(1)$-invariant point with Hamiltonian $H(\frac{\pi}{2}) = H_1$. Even on the lattice, this point has remarkable properties: an exact lattice supersymmetry [13], and an Onsager-algebra symmetry (our Hamiltonian $H_1$ here is $-H_0$ of [12]). The continuum limit is a supersymmetric CFT [19,20] that can be written in terms of a product of a free-boson and free-fermion theories, so that $c = \frac{3}{2} = 1 + \frac{1}{2}$. The toroidal partition function is $Z_{\text{s-a}}(\sqrt{3})$ in the notation of [21], where the $\sqrt{3}$ is the radius of the boson [13]. An orbifold couples the two CFTs by imposing certain selection rules for the states, but otherwise the boson and fermion theories are independent.

This particular CFT has the remarkable property that it can be split up into a product of two CFTs in two ways [21]: it also is a product of the three-state Potts and the tricritical Ising (TCI) CFTs ($c = \frac{3}{2} = \frac{4}{5} + \frac{7}{10}$). As with the boson-fermion decomposition, the Potts and TCI theories are independent except for selection rules. Each scaling dimension of each operator in this $c = \frac{3}{2}$ CFT therefore can be split up in two ways:

$$\Delta_a = \Delta_{a,\text{B}} + \Delta_{a,\text{F}} = \Delta_{a,P} + \Delta_{a,\text{TCI}} , \qquad (11)$$

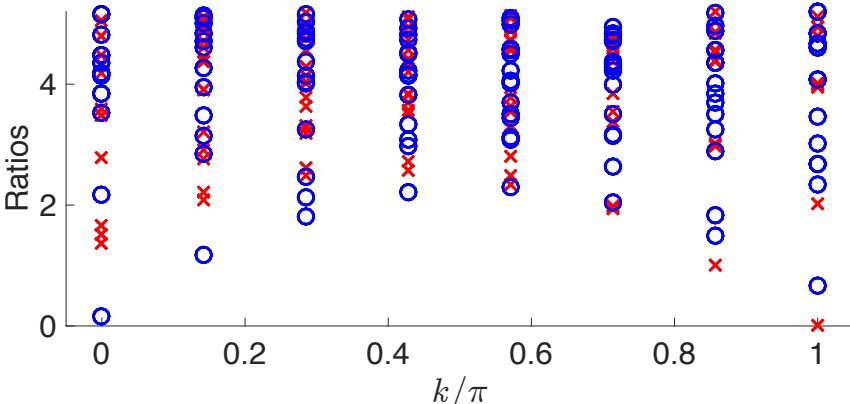

Figure 4: The ratio (10) with $E_b = E_d = E_0^0$ and $E_c = E_0^1$ for the integrable antiferromagnetic Potts Hamiltonian $H(\pi) = -H_P$ for many levels found using exact diagonalization at $L=14$. Red crosses have $\omega^Q = 1$, and blue circles have $\omega^Q = \omega$. The levels are plotted as a function of momentum $k$, with the ground state having $k = \pi$. The self-dual dimension-$\frac{3}{2}$ operator is one of the lowest-lying crosses at $k = 0$.

where e.g. $\Delta_{a,\mathrm{B}}$ is a scaling dimension in the free-boson theory. The remarkable properties of this CFT go even deeper than (11). The energy-momentum tensor of any of the component CFTs can be expressed as linear combinations of three dimension-2 operators in the other theory [40, 41]. The three are both the energy-momentum tensors and a third operator, with dimensions $(\Delta_{\mathrm{B}}, \Delta_{\mathrm{F}}) = (\frac{3}{2}, \frac{1}{2})$, or $(\Delta_P, \Delta_{\mathrm{TCI}}) = (\frac{3}{5}, \frac{7}{5})$. These linear expressions make it straightforward to relate certain primary fields in the Potts and TCI CFTs to free fermions and bosons.

The question now is what happens when $\theta$ is taken away from $\frac{\pi}{2}$, and so $H_P$ is added to the Hamiltonian. We give in Appendix A a list of all relevant and marginal operators with their symmetry properties. This list follows from using the partition functions presented in Ref. [21] along with a careful analysis of the discrete symmetries. We find that none of the relevant operators in the $c = \frac{3}{2}$ CFT are both self-dual and preserve all the symmetries of $H(\theta)$. For example, the self-dual dimension-7/8 operators have non-zero momentum and so violate translation symmetry. As indicated above and apparent in the table, however, there are three marginal operators of scaling dimension 2. These operators cannot be marginally relevant, as they all have conformal spin $\pm 2$, which cannot renormalize. Thus at most they are exactly marginal.

To proceed further, we must do numerics. We find that indeed $H(\theta)$ remains critical for a large region as $\theta$ is varied from $\frac{\pi}{2}$. Namely, exact diagonalization indicates that energy differences of low-lying states remain proportional to $1/L$, as in a CFT. Moreover, our DMRG calculation of entanglement-entropy scaling [35] in this region remains consistent with that in a $c = \frac{3}{2}$ CFT. The criticality therefore extends to a full phase, labeled as "$c$"$= \frac{4}{5} + \frac{7}{10}$ in the top part of Figure 1.

We gave this phase an unusual name for the following reasons. Even though the universal term in the entanglement entropy remains constant throughout the phase, the spectrum changes. We find the scaling dimensions no longer obey (11) for $\theta \neq \frac{\pi}{2}$, but to reasonably good numerical accuracy instead obey

$$\Delta_a(\theta) = v_P(\theta)\Delta_{a,P} + v_{\mathrm{TCI}}(\theta)\Delta_{a,\mathrm{TCI}} , \tag{12}$$

where, crucially, the ratio of "Fermi velocities" $v_{\mathrm{TCI}}/v_P$ *does not depend on the level* $a$. The data cannot be fit well by using different fermi and boson velocities, but only by those for tricritical

Ising and Potts sectors. We give a plot of the TCI Fermi velocity $v_{r,k}^j$ for the $j^{\text{th}}$ excited state in the sector with $\mathbb{Z}_3$ charge $\omega^r$ and momentum $k$ in Figure 5, setting $v_P = 1$. We extract its value from (12) by first determining the energies $E_{r,k}^j$ using exact diagonalization for even $L$ from 6 through 16, and then fitting to a form $E_{r,k}^j/L + B/L^2$. We then extract the scaling dimensions using (10) to eliminate non-universal quantities. An important caveat is that to obtain (12), we considered only levels not degenerate at $\theta = \frac{\pi}{2}$, as degenerate ones have a more complicated mixing.

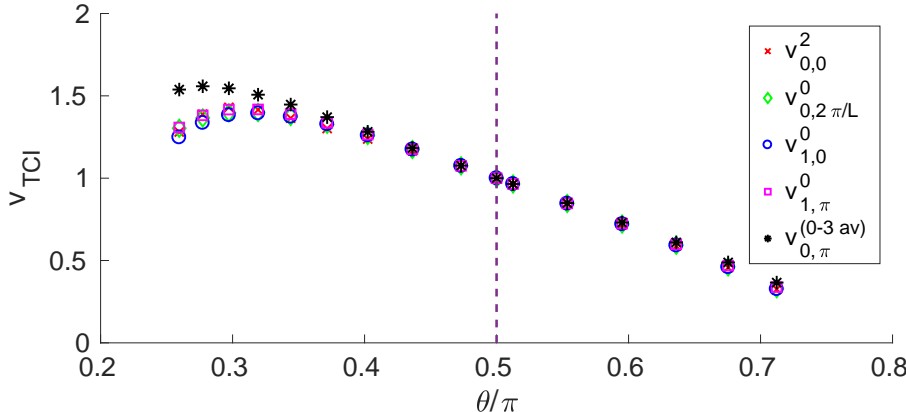

Figure 5: The ratio of "Fermi velocities" $v_{\text{TCI}}/v_P$ vs. $\theta$ for levels with $\Delta_a = \dots$ at $\theta = \frac{\pi}{2}$, determined from (12) as explained in the text. The $v_{0,\pi}^{(0\text{-}3\text{ av})}$ is the average of the four lowest levels in this sector, which are degenerate at the integrable point.

If there were only a single exactly marginal operator, (12) would be exact, and the decomposition into two CFTs

$$H(\theta) \rightarrow v_P(\theta)\mathcal{H}_P + v_{\text{TCI}}(\theta)\mathcal{H}_{\text{TCI}} \tag{13}$$

presumably would hold throughout this critical region. The presence of three self-dual and symmetry preserving operators at $\theta = \frac{\pi}{2}$, however, gives *two* marginal perturbations. Changing the relative fermi velocities is one of these exactly marginal perturbations, so the open question here is the role of the other marginal perturbation. Since one of these operators has dimension $(\Delta_{\text{TCI}}, \Delta_P) = (\frac{3}{5}, \frac{7}{5})$, it couples the tricritical Ising and Potts theories. If it remains exactly marginal after perturbation, (12) will only be approximate, as the data at the extremes of Figure 5 suggest. For this reason, we included the quotes in the "c"$= \frac{4}{5} + \frac{7}{10}$ denoting this critical region in Figure 1. However, it is entirely possible that the variations in the data are merely finite-size effects increasing as the phase transitions are approached, so that the coupling operator is marginally irrelevant and (5) is exact. Indeed, such a marginally irrelevant perturbation occurs both in a $c = 3/2$ field theory [42] and a lattice model [43], also resulting in a Lorentz-symmetry-breaking perturbation. There, however, the effect is to restore Lorentz symmetry at large distances, whereas here the effect would be to leave the two effective theories decoupled.

The transitions out of this fourth large critical region both seem to be to gapless incommensurate phases, as ground-state level-crossings occur in exact diagonalization. Increasing $\theta$ toward the antiferromagnetic Potts phase, the Fermi velocity $v_{\text{TCI}}$ in Figure 5 quite clearly is vanishing, indicating another interesting phase transition. Because of the preponderance of low-lying energy levels, this transition unfortunately is rather difficult to analyze numerically. A gapless incommensurate phase seems to describe the region from $\theta \approx 0.87\pi$ to $\theta \approx 0.93\pi$. As $\theta$ is decreased, a small gapless incommensurate region also seems to intervene before the gapped order-disorder coexistence phase is reached.

# 4  Gapped order-disorder coexistence

Both Potts phases terminate at one end in a tricritical Potts point. Changing $\theta$ away from $\theta_{\text{TCP}}$ or $\theta_{\text{TCP}'}$ gives a relevant perturbation by an operator of dimension $\frac{10}{7}$. As opposed to the behavior at the $c=1$ point, the effective field theories are not the same for both signs of perturbation. In one direction, the RG flow goes back to the ferromagnetic $c = \frac{4}{5}$ critical point [26]. As we described above, this ensuing Potts 1 and Potts 2 critical phases, the former separates the duality-broken Potts ordered and disordered phases, and the latter separating the not-$A$ and RSPT phases [4].

Here we consider the phases found by going away from the tricritical Potts points in the other direction. When the perturbation has the other sign, the self-dual line remains the transition line, but the model is gapped and the transition first-order [26]. The ensuing effective field theory describing this region is integrable and massive [29]. The TCP points thus separate the first and second-order lines, providing a natural generalization of the familiar physics of the tricritical Ising model. The disordered and three ordered ground states coexist along these first-order lines, resulting in a fractional supersymmetry we describe below.

We first establish the order-disorder coexistence on the lattice rigorously at a special frustration-free point where the multiple ground states can be found exactly, just as in the $\mathbb{Z}_2$ case [23]. Here this point is at $\lambda_1 = \lambda_{\text{P}}/3 > 0$, where $\theta = \theta_{\text{ff}} \approx 0.102\pi$. The four ground states are

$$|000\cdots0\rangle, \quad |111\cdots1\rangle, \quad |222\cdots2\rangle, \quad |\hat{0}\hat{0}\hat{0}\cdots\hat{0}\rangle, \tag{14}$$

where $\sigma|A\rangle = \omega^A|A\rangle$ for $A = 0, 1, 2$, while $|\hat{0}\rangle \equiv (|0\rangle + |1\rangle + |2\rangle)/\sqrt{3}$, which obeys $\tau|\hat{0}\rangle = |\hat{0}\rangle$. In the $\sigma$-diagonal basis, the first three ground states are completely ordered while the last is the equal-amplitude sum over all states. The latter ground state is dual to the other three, as hinted at by the fact that is a product state in the $\tau_j$-diagonal basis.

To prove that the states (14) are the ground states at the frustration-free point, we write the corresponding Hamiltonian $H(\theta_{\text{ff}})$ as a sum over projectors. There are two projectors for each pair of nearest-neighbor sites, so that

$$H(\theta_{\text{ff}}) = -4L + 6\sum_{j=1}^{L}\left(P_j^{(1)} + P_j^{(2)}\right), \tag{15}$$

where $(P_j^{(r)})^2 = P_j^{(r)}$ for $r = 1, 2$. Explicit expressions for these projectors are easiest to write out in the $\tau_j$-diagonal basis, where $\tau|\hat{A}\rangle = \omega^A|\hat{A}\rangle$ for $A = 0, 1, 2$. Acting on the sites $j, j+1$ they are

$$2P_j^{(1)} = \left(|\hat{1}\hat{0}\rangle - |\hat{2}\hat{2}\rangle\right)\left(\langle\hat{1}\hat{0}| - \langle\hat{2}\hat{2}|\right) + \left(|\hat{2}\hat{0}\rangle - |\hat{1}\hat{1}\rangle\right)\left(\langle\hat{2}\hat{0}| - \langle\hat{1}\hat{1}|\right) + \left(|\hat{1}\hat{2}\rangle - |\hat{2}\hat{1}\rangle\right)\left(\langle\hat{1}\hat{2}| - \langle\hat{2}\hat{1}|\right);$$

$$2P_j^{(2)} = \left(|\hat{0}\hat{1}\rangle - |\hat{2}\hat{2}\rangle\right)\left(\langle\hat{0}\hat{1}| - \langle\hat{2}\hat{2}|\right) + \left(|\hat{0}\hat{2}\rangle - |\hat{1}\hat{1}\rangle\right)\left(\langle\hat{0}\hat{2}| - \langle\hat{1}\hat{1}|\right) + \left(|\hat{1}\hat{2}\rangle - |\hat{2}\hat{1}\rangle\right)\left(\langle\hat{1}\hat{2}| - \langle\hat{2}\hat{1}|\right).$$

Expressions of these operators in terms of the $\sigma_j$ and $\tau_j$ can be found in appendix B. These expressions show immediately that $|\hat{0}\hat{0}...\hat{0}\rangle$ is annihilated by all the two-site projectors $P_j^{(r)}$, and so must be a ground state of the Hamiltonian with energy $-4L$. A few more minutes of additional work shows that $|AAA...A\rangle$ is annihilated by all of them as well, and so also are ground states. Indeed, using the operator given in [5] shows that duality maps any of the latter three to $|\hat{0}\hat{0}...\hat{0}\rangle$ (recall duality is not invertible). These four states are the only ground states for $L \geq 3$, as it is straightforward to verify that these are the only states annihilated by all projectors. Analogous frustration-free points for the $Q$-state Potts model with a nearest-neighbor $S_Q$ preserving perturbation can be found by using the Temperley–Lieb formulation

[8,44]. These have $Q + 1$ degenerate ground states, $Q$ of which are completely ordered and one completely disordered.

Our numerics confirm that order-disorder coexistence persists throughout a gapped phase for $\theta$ on both sides of $\theta_{\text{ff}}$. The self-duality makes this coexistence natural, as any ordered ground state will map to a disordered one under the duality. The self-duality also gives the exact location of these lines in the bigger parameter space, if not the location of the tricritical point itself. The physics thus generalizes that of the $\mathbb{Z}_2$ case [22,23]. Moreover, past the frustration-free point it contains an incommensurate length scale, as in the analogous phase surrounding the Majumdar–Ghosh point in a frustrated $su(2)$-invariant antiferromagnet [45, 46]. Namely, for $\theta > \theta_{\text{ff}}$, level crossings occur amongst excited states, and the correlators exhibit oscillations on top of the exponential decay. These oscillations are readily apparent in $\langle \sigma_i^\dagger \sigma_j \rangle$ plotted in figure 6. $\theta$ is increased further, the oscillations persist. As the oscillations are rather small in magnitude (note the $y$-axis values on the log plot) we were unable to determine the period precisely. As best as we can tell, a gapless incommensurate phase then occurs as a result of a level crossing the ground state at $\theta \sim 0.16\pi$.

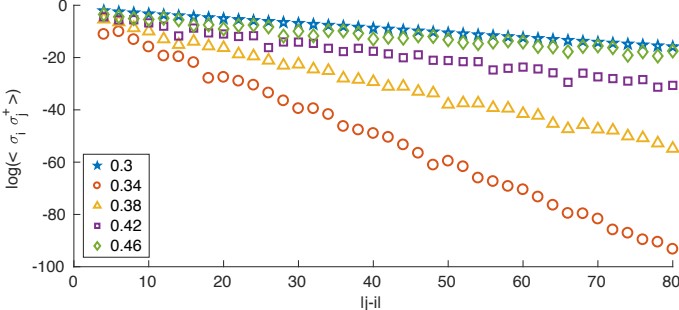

Figure 6: The log of the spin-spin correlator in the ground state in the gapped phase at $\lambda_{\text{P}} = 1$ and various $\lambda_1$ values given in the legend. The correlator was obtained using DMRG with $L = 300$ and a maximum bond dimension of 400. The incommensurability is readily apparent in the oscillations present for $\lambda_1 > 1/3$.

Since the second Potts phase terminates in the TCP point at the bottom left of Figure 1, universality arguments imply also coexistence between the three ground states of the not-$A$ phase coexist and the unique one of the SPT phase. A direct lattice derivation of this phase is difficult, as no frustration-free point occurs here. Our DMRG numerics indicate a large correlation length with substantial oscillations, but the ground states we find have a fairly small bond dimension. We take the latter as a strong sign of the existence of a gap in at least a small region. It appears that the gapped region terminates at $\theta \approx -0.70\pi$ with a small incommensurate phase in the region before the critical antiferromagnetic phase starting at $\theta \approx -0.73\pi$.

The order-disorder coexistence phase also has a very intriguing emergent "fractional supersymmetry", generalizing the emergent supersymmetry in the analogous $\mathbb{Z}_2$ phase. Supersymmetric Hamiltonians generically can be written as (sums over) squares of fermionic supersymmetry generators [47], and in the exact-scattering-matrix approach, appropriate fermionic operators $\mathcal{Q}_L$ and $\mathcal{Q}_R$ indeed can be defined so that $H_2 = \mathcal{Q}_L^2 + \mathcal{Q}_R^2$ [29]. Remarkably, this approach can be extended to the region around the TCP point, a consequence of conformal spin $\pm 4/3$ operators [27] remaining symmetry generators. The effective Hamiltonian of the $S_3$-invariant order-disorder coexistence phase thus can be written as the sum of two *cubes* of parafermionic operators, i.e. $H_3 = Q_L^3 + \overline{Q}_R^3$ [29,48].

In our earlier work [23] we showed that in the critical Ising chain with a particular self-dual perturbation, the lattice Hamiltonian can be written as $H_2 = Q^{+2} + Q^{-2}$. The fermionic $Q^\pm$ are

sums over products of odd numbers of Majorana fermions on neighboring sites. Although they do not commute with the Hamiltonian on the lattice, numerics indicated they renormalize onto the supersymmetry generators in the scaling limit. The natural generalization of this construction to our 3-state model uses *parafermions* $\psi_a$ instead of Majorana fermions. They are defined by

$$\psi_{2j-1} = \sigma_j \prod_{k<j} \tau_k^x, \qquad \psi_{2j} = \omega \sigma_j \tau_j \prod_{k<j} \tau_k, \tag{16}$$

so that e.g. $\tau_j = \omega^2 \psi_{2j-1} \psi_{2j}$ and $\sigma_j^\dagger \sigma_{j+1} = \omega^2 \psi_{2j} \psi_{2j+1}$. They obey the algebra

$$\psi_a^\dagger = \psi_a^2, \qquad \psi_a^3 = 1, \qquad \omega^Q \psi_a = \omega \psi_a \omega^Q, \qquad \psi_a \psi_b = \omega \psi_b \psi_a \text{ for } a < b.$$

One nice feature of the parafermion operators is their nice behavior under the duality (5), transforming as $\psi_a \to \psi_{a+1}$. Another fact worth noting is that when $H(\theta)$ is written in terms of the $\psi_a$, the farthest-apart terms are $\psi_a^\dagger \psi_{a+2}$ and $\psi_a \psi_{a+1} \psi_{a+2}$ and their Hermitian conjugates.

The simplest lattice parafermion operator $\mathcal{Q}$ giving something non-trivial when cubed is

$$\mathcal{Q} = \sum_a \left( \alpha \psi_a + \beta \psi_a^\dagger \psi_{a+1}^\dagger \right). \tag{17}$$

Such a $Q$ is not Hermitian. Requiring charge conjugation, parity and time-reversal fixes $\beta^3 \in \mathbb{R}$ and $\alpha = 2\omega^2 \beta$. Then some straightforward but tedious algebra yields

$$\mathcal{Q}^3 + \mathcal{Q}^{\dagger 3} = H(\theta_{\text{ff}}) - \sum_{j=1}^{L} \left[ \tau_j \tau_{j+1}^\dagger + \tau_j^\dagger \tau_{j+1} + \sigma_j \sigma_{j+1} \sigma_{j+2} + \sigma_j^\dagger \sigma_{j+1}^\dagger \sigma_{j+2}^\dagger \right], \tag{18}$$

where $3\beta^3 = 1$. The terms inside the square brackets are longer-range, in the sense that they involve products like $\psi_a \psi_{a+1}^\dagger \psi_{a+2}^\dagger \psi_{a+3}$. This expression strongly suggests that the parafemionic operators (17) provide lattice analogs of the fractional supersymmetry generators.

We can remove the extra terms in (18) and extend the results to all $\theta$ by considering a sum over generators as in the $\mathbb{Z}_2$ case. We then define

$$Q_n = \sum_j \left( \alpha_{n,a} \psi_a + \beta_{n,a} \psi_a^\dagger \psi_{a+1}^\dagger \right). \tag{19}$$

We show in appendix B that coefficients $\alpha_{n,a}$ and $\beta_{n,a}$ can be found so that

$$H(\theta) = Q_1^3 + Q_1^{\dagger 3} + Q_2^3 + Q_2^{\dagger 3} + Q_3^3 + Q_3^{\dagger 3} \tag{20}$$

for any $\theta$. The precise meaning and consequences of (20) are not immediately apparent to us, but it does seem rather natural in light of the emergent symmetries described in [29, 48].

# 5 Conclusion

We have found the phase diagram of the one-dimensional self-dual 3-state Potts model perturbed by the only self-dual nearest-neighbor interaction obeying all of its symmetries. Two critical Potts phases appear, separated by a $U(1)$-invariant critical point. One Potts line separates novel RSPT and not-*A* phases [4], the other the usual ordered and disordered phases. The antiferromagnetic Potts critical point extends to a full phase here, as opposed to the corresponding square-lattice antiferromagnet. Even more striking is finding an unusual "c"$=\frac{4}{5} + \frac{7}{10}$

phase, where another $U(1)$-invariant critical point splits into CFTs with different fermi velocities. At least one and probably two gapped phases with $S_3$ order-disorder coexistence occur as well, separated from Potts phases by a tricritical point.

Several of our results are rather striking, and would be well worth additional study. The splitting via distinct fermi velocities occurring in the "c"=$\frac{4}{5} + \frac{7}{10}$ phase is rather unusual, especially given that the component theories are strongly interacting. In particular, it would be nice to know whether these two CFTs are decoupled or are interacting. If the latter, how does one write down such interactions in field theory? To understand how the decoupling works (or doesn't), developing a RG analysis in the fashion of [42, 43] likely would be illuminating, as also would be writing lattice analogs of various operators. At a more formal level, the decoupling even at the $c = \frac{3}{2}$ point is very interesting, as it leads to being able to derive marvelous explicit expressions for CFT correlators, for example that given in [49]. Concerning the gapped phases, we also know of no other lattice models where Hamiltonians can be written as a sum over the cubes of parafermionic operators. Connecting it to field theory in a more transparent way would be very desirable.

# Acknowledgements

We would like to thank Eduardo Fradkin, Yichen Hu and Eric Vernier for useful discussions. This work was supported by EPSRC through grant EP/N509711/1 1734484 (EOB) along with grants EP/S020527/1 and EP/N01930X (PF).

# A  The $c = \frac{3}{2} = 1 + \frac{1}{2} = \frac{4}{5} + \frac{7}{10}$ CFT

In Table 1 we present a list of all marginal and relevant operators in the CFT describing the continuum limit of $H(\frac{\pi}{2}) = H_1$. The dimension $\Delta$ of each is given in both forms (11), with the additional splitting of each into left and right components $(\Delta_L, \Delta_R)$, so that e.g. the dimension-$\frac{5}{24}$ operator in the second row is of dimension $\frac{3}{40} = \frac{3}{80} + \frac{3}{80}$ in the tricritical Ising CFT and $\frac{2}{15} = \frac{1}{15} + \frac{1}{15}$ in the three-state Potts CFT. A primed dimension denotes the Virasoro raising operator $L_{-1}$ or $\bar{L}_{-1}$ acting on the primary field of that dimension, while a double-primed number indicates the action of $L_{-2}$ or $\bar{L}_{-2}$. Thus, for example, $0''$ is the energy-momentum tensor.

We then list their symmetry charges of the fields. The conformal spin is $s = \Delta_L - \Delta_R$, the $\mathbb{Z}_3$ charge is $\omega^r$, and the lattice momentum is $k$. The "electric" and "magnetic" charges $[m, n]$ are those under the two $U(1)$ symmetries $Q$ and $\widehat{Q}$ respectively, so that $\omega^r = \omega^m$. For all the states with $r = 0$, we give the eigenvalues under duality $\mathcal{D}$ and the product of it with parity: $\mathcal{D}' = \mathcal{D}\mathcal{P}$. The reason for the restriction is that when $r \neq 0$, duality maps periodic boundary conditions to twisted sectors and so does not have a well-defined eigenvalue. In order to simplify the table, we have not written the TCI+P states as parity eigenstates, but they can be found simply by taking appropriate combinations of the states with left and right exchanged. The duality eigenvalues therefore apply only to the B+F expressions.

A few comments on these symmetry properties are in order. The $\mathbb{Z}_3$ symmetry lives solely in the three-state Potts sector in the TCI + P picture, and in the boson in the B+F picture, as it is generated by $\omega^Q$. The two $(1/15, 1/15)$ operators in Potts have $r = \pm 1$, as do the two $(2/3, 2/3)$ fields, while all others have $r = 0$. The lattice momentum sectors $k = 0$ and $k = \pi$ are determined solely by the corresponding $\mathbb{Z}_2$ sectors in the TCI CFT (see [50] for how that works), while they are given by $\pi$ times $(m + 2n) \bmod 2$ in the B + F picture. We find that in TCI + P, duality is just given by the Potts duality. The mapping of operators between

Table 1: The relevant and marginal operators for $H(\frac{\pi}{2}) = H_1$. The precise definitions of the scaling dimensions and the charges are given in the text. A primed dimension denotes the Virasoro raising operator $L_{-1}$ or $\bar{L}_{-1}$ acting on the primary field of that dimension, while a double-primed number indicates the action of $L_{-2}$ or $\bar{L}_{-2}$. The linear combinations of the bosonic and fermion operators are chosen to behave nicely under duality and parity. For operators with charge $r = 1$, the corresponding operator with $r = -1$ is not given. For fields with $k \neq 0, \pi$, the corresponding operator with momentum $-k$ not given.

| $\Delta$ | $s$ | $r$ | $k$ | TCI + P | B + F | $[m,n]$ | $\mathcal{D}$ | $\mathcal{D}'$ |
|---|---|---|---|---|---|---|---|---|
| $0$ | $0$ | $0$ | $0$ | $(0+0, 0+0)$ | $(0+0, 0+0)$ | $[0,0]$ | $+1$ | $+1$ |
| $\frac{5}{24}$ | $0$ | $1$ | $\pi$ | $\left(\frac{3}{80}+\frac{1}{15}, \frac{3}{80}+\frac{1}{15}\right)$ | $\left(\frac{1}{24}+\frac{1}{16}, \frac{1}{24}+\frac{1}{16}\right)$ | $[1,0]$ | - | - |
| $\frac{1}{3}$ | $0$ | $1$ | $0$ | $\left(\frac{1}{10}+\frac{1}{15}, \frac{1}{10}+\frac{1}{15}\right)$ | $\left(\frac{1}{6}+0, \frac{1}{6}+0\right)$ | $[-2,0]$ | - | - |
| $\frac{7}{8}$ | $0$ | $0$ | $\pi$ | $\left(\frac{7}{16}+0, \frac{7}{16}+0\right)$ | $\left(\frac{3}{8}+\frac{1}{16}, \frac{3}{8}+\frac{1}{16}\right)$ | $[3,0]+[0,\frac{1}{2}]+[-3,0]+[0,-\frac{1}{2}]$ | $+1$ | $+1$ |
| | | | | $\left(\frac{7}{16}+0, \frac{3}{80}+\frac{2}{5}\right)$ | $\left(\frac{3}{8}+\frac{1}{16}, \frac{3}{8}+\frac{1}{16}\right)$ | $[3,0]-[0,\frac{1}{2}]-[-3,0]+[0,-\frac{1}{2}]$ | $+1$ | $-1$ |
| | | | | $\left(\frac{3}{80}+\frac{2}{5}, \frac{7}{16}+0\right)$ | $\left(\frac{3}{8}+\frac{1}{16}, \frac{3}{8}+\frac{1}{16}\right)$ | $[3,0]+[0,\frac{1}{2}]-[-3,0]-[0,-\frac{1}{2}]$ | $-1$ | $+1$ |
| | | | | $\left(\frac{3}{80}+\frac{2}{5}, \frac{3}{80}+\frac{2}{5}\right)$ | $\left(\frac{3}{8}+\frac{1}{16}, \frac{3}{8}+\frac{1}{16}\right)$ | $[3,0]-[0,\frac{1}{2}]+[-3,0]-[0,-\frac{1}{2}]$ | $-1$ | $-1$ |
| $1$ | $0$ | $0$ | $0$ | $\left(\frac{1}{10}+\frac{2}{5}, \frac{1}{10}+\frac{2}{5}\right)$ | $\left(0+\frac{1}{2}, 0+\frac{1}{2}\right)$ | $[0,0]$ | $-1$ | $-1$ |
| $1$ | $1$ | $0$ | $\frac{2\pi}{L}$ | $\left(\frac{3}{5}+\frac{2}{5}, 0+0\right)$ | $(1+0, 0+0)$ | $[0,0]$ | $-1$ | $+1$ |
| $\frac{29}{24}$ | $1$ | $1$ | $\pi+\frac{2\pi}{L}$ | $\left(\frac{7}{16}+\frac{2}{3}, \frac{3}{80}+\frac{1}{15}\right)$ | $\left(\frac{1}{24}'+\frac{1}{16}, \frac{1}{24}+\frac{1}{16}\right)$ | $[1,0]$ | - | - |
| | | | | $\left(\frac{3}{80}'+\frac{1}{15}, \frac{3}{80}+\frac{1}{15}\right)$ | $\left(\frac{1}{24}+\frac{1}{16}', \frac{1}{24}+\frac{1}{16}\right)$ | $[1,0]$ | - | - |
| | | | | $\left(\frac{3}{80}+\frac{1}{15}', \frac{3}{80}+\frac{1}{15}\right)$ | $\left(\frac{25}{24}+\frac{1}{16}, \frac{1}{24}+\frac{1}{16}\right)$ | $[-2,-\frac{1}{2}]$ | - | - |
| $\frac{4}{3}$ | $0$ | $1$ | $0$ | $\left(0+\frac{2}{3}, 0+\frac{2}{3}\right)$ | $\left(\frac{1}{6}+\frac{1}{2}, \frac{1}{6}+\frac{1}{2}\right)$ | $[-2,0]$ | - | - |
| | | | | $\left(\frac{3}{5}+\frac{1}{15}, 0+\frac{2}{3}\right)$ | $\left(\frac{1}{6}+\frac{1}{2}, \frac{2}{3}+0\right)$ | $[1,-\frac{1}{2}]$ | - | - |
| | | | | $\left(0+\frac{2}{3}, \frac{3}{5}+\frac{1}{15}\right)$ | $\left(\frac{2}{3}+0, \frac{1}{6}+\frac{1}{2}\right)$ | $[1,\frac{1}{2}]$ | - | - |
| | | | | $\left(\frac{3}{5}+\frac{1}{15}, \frac{3}{5}+\frac{1}{15}\right)$ | $\left(\frac{2}{3}+0, \frac{2}{3}+0\right)$ | $[4,0]$ | - | - |
| $\frac{15}{8}$ | $1$ | $0$ | $\pi+\frac{2\pi}{L}$ | $\left(\frac{7}{16}'+0, \frac{7}{16}+0\right)$ | $\left(\frac{3}{8}'+\frac{1}{16}, \frac{3}{8}+\frac{1}{16}\right)$ | $[3,0]+[0,\frac{1}{2}]+[-3,0]+[0,-\frac{1}{2}]$ | $+1$ | $+1$ |
| | | | | $\left(\frac{3}{80}+\frac{7}{5}, \frac{7}{16}+0\right)$ | $\left(\frac{3}{8}+\frac{1}{16}', \frac{3}{8}+\frac{1}{16}\right)$ | $[3,0]+[0,\frac{1}{2}]+[-3,0]+[0,-\frac{1}{2}]$ | $+1$ | $+1$ |
| | | | | $\left(\frac{7}{16}'+0, \frac{3}{80}+\frac{2}{5}\right)$ | $\left(\frac{3}{8}'+\frac{1}{16}, \frac{3}{8}+\frac{1}{16}\right)$ | $[3,0]-[0,\frac{1}{2}]-[-3,0]+[0,-\frac{1}{2}]$ | $+1$ | $-1$ |
| | | | | $\left(\frac{3}{80}+\frac{7}{5}, \frac{3}{80}+\frac{2}{5}\right)$ | $\left(\frac{3}{8}+\frac{1}{16}', \frac{3}{8}+\frac{1}{16}\right)$ | $[3,0]-[0,\frac{1}{2}]-[-3,0]+[0,-\frac{1}{2}]$ | $+1$ | $-1$ |
| | | | | $\left(\frac{3}{80}+\frac{2}{5}', \frac{7}{16}+0\right)$ | $\left(\frac{3}{8}'+\frac{1}{16}, \frac{3}{8}+\frac{1}{16}\right)$ | $[3,0]+[0,\frac{1}{2}]-[-3,0]-[0,-\frac{1}{2}]$ | $-1$ | $+1$ |
| | | | | $\left(\frac{3}{80}'+\frac{2}{5}, \frac{7}{16}+0\right)$ | $\left(\frac{3}{8}+\frac{1}{16}', \frac{3}{8}+\frac{1}{16}\right)$ | $[3,0]+[0,\frac{1}{2}]-[-3,0]-[0,-\frac{1}{2}]$ | $-1$ | $+1$ |
| | | | | $\left(\frac{3}{80}+\frac{2}{5}', \frac{3}{80}+\frac{2}{5}\right)$ | $\left(\frac{3}{8}'+\frac{1}{16}, \frac{3}{8}+\frac{1}{16}\right)$ | $[3,0]-[0,\frac{1}{2}]+[-3,0]-[0,-\frac{1}{2}]$ | $-1$ | $-1$ |
| | | | | $\left(\frac{3}{80}'+\frac{2}{5}, \frac{3}{80}+\frac{2}{5}\right)$ | $\left(\frac{3}{8}+\frac{1}{16}', \frac{3}{8}+\frac{1}{16}\right)$ | $[3,0]-[0,\frac{1}{2}]+[-3,0]-[0,-\frac{1}{2}]$ | $-1$ | $-1$ |
| $2$ | $0$ | $0$ | $0$ | $\left(\frac{3}{5}+\frac{2}{5}, \frac{3}{5}+\frac{2}{5}\right)$ | $(1+0, 1+0)$ | $[0,0]$ | $-1$ | $-1$ |
| $2$ | $1$ | $0$ | $\frac{2\pi}{L}$ | $\left(\frac{3}{2}+0, \frac{1}{10}+\frac{2}{5}\right)$ | $\left(\frac{3}{2}+0, 0+\frac{1}{2}\right)$ | $[3,\frac{1}{2}]+[-3,-\frac{1}{2}]$ | $+1$ | $-1$ |
| | | | | $\left(\frac{1}{10}+\frac{7}{5}, \frac{1}{10}+\frac{2}{5}\right)$ | $\left(1+\frac{1}{2}, 0+\frac{1}{2}\right)$ | $[0,0]$ | $+1$ | $-1$ |
| | | | | $\left(\frac{1}{10}+\frac{2}{5}', \frac{1}{10}+\frac{2}{5}\right)$ | $\left(\frac{3}{2}+0, 0+\frac{1}{2}\right)$ | $[3,\frac{1}{2}]-[-3,-\frac{1}{2}]$ | $-1$ | $-1$ |
| | | | | $\left(\frac{1}{10}'+\frac{2}{5}, \frac{1}{10}+\frac{2}{5}\right)$ | $\left(0+\frac{1}{2}', 0+\frac{1}{2}\right)$ | $[0,0]$ | $-1$ | $-1$ |
| $2$ | $2$ | $0$ | $\frac{4\pi}{L}$ | $(0''+0, 0+0)$ | $(0''+0, 0+0)$ | $[0,0]$ | $+1$ | $+1$ |
| | | | | $(0+0'', 0+0)$ | $(0+0'', 0+0)$ | $[0,0]$ | $+1$ | $+1$ |
| | | | | $\left(\frac{3}{5}+\frac{7}{5}, 0+0\right)$ | $\left(\frac{3}{2}+\frac{1}{2}, 0+0\right)$ | $[3,\frac{1}{2}]-[-3,-\frac{1}{2}]$ | $+1$ | $+1$ |
| | | | | $\left(\frac{3}{5}'+\frac{2}{5}, 0+0\right)$ | $\left(\frac{3}{2}+\frac{1}{2}, 0+0\right)$ | $[3,\frac{1}{2}]+[-3,-\frac{1}{2}]$ | $-1$ | $+1$ |
| | | | | $\left(\frac{3}{5}+\frac{2}{5}', 0+0\right)$ | $(1'+0, 0+0)$ | $[0,0]$ | $-1$ | $+1$ |

TCI+P and B+F descriptions then requires that duality act on both the boson and fermion theories. We define $\mathcal{D}$ to send $m \leftrightarrow -n$ and $(0,1/2) \to (0,1/2)$, while $\mathcal{D}'$ gives $m \leftrightarrow n$ and $(0,1/2) \to -(0,1/2)$.

Going through the table, one sees that all relevant operators have non-vanishing charge under at least one of the symmetries, or are not self-dual. Only three marginal operators are self-dual and invariant under all symmetries including parity. Each of these is the sum of an operator of scaling dimension $(2,0)$ and its parity conjugate of dimension $(0,2)$. In the B+F language, these are found from the boson stress-energy tensor $(0''+0, 0+0)$, the fermion stress-energy tensor $(0+0'', 0+0)$, along with the third field $(3/2+1/2, 0+0)$ (plus their conjugates). The latter operator is the reason why the model may not split exactly to a combination of the Potts and TCI models with different Fermi velocities, as explained in section 3.

## B The Hamiltonian as a sum of cubes

Here we show how to write the Hamiltonian (8) as the sum of cubes of parafermionic operators given in (20). A useful expression in the analysis and in the derviation of the form (18) for the frustration-free point is

$$
\begin{aligned}
2 - 6P_j^{(1)} &= \sigma_j^\dagger \sigma_{j+1} + \sigma_j \sigma_{j+1}^\dagger + \tau_j + \tau_j^\dagger + \omega^2 \tau_j \sigma_j \sigma_{j+1}^\dagger \\
&\quad + \omega \tau_j \sigma_j^\dagger \sigma_{j+1} + \omega \tau_j^\dagger \sigma_j \sigma_{j+1}^\dagger + \omega^2 \tau_j^\dagger \sigma_j^\dagger \sigma_{j+1}, \\
2 - 6P_j^{(2)} &= \sigma_j^\dagger \sigma_{j+1} + \sigma_j \sigma_{j+1}^\dagger + \tau_{j+1} + \tau_{j+1}^\dagger \\
&\quad + \omega \sigma_j^\dagger \sigma_{j+1} \tau_{j+1} + \omega^2 \sigma_j^\dagger \sigma_{j+1} \tau_{j+1}^\dagger + \omega^2 \sigma_j \sigma_{j+1}^\dagger \tau_{j+1} + \omega \sigma_j \sigma_{j+1}^\dagger \tau_{j+1}^\dagger.
\end{aligned}
$$

Plugging (19) into (20) turns out to give eight equations for eight unknowns. Parametrizing the unknowns via

$$
\alpha_{n,a} = \alpha e^{i\theta_n} e^{\frac{2\pi n a i}{3}}, \qquad \beta_{n,a} = \beta e^{i\phi_n} e^{\frac{2\pi n a i}{3}}, \qquad \mu_n = \theta_n + 2\phi_n
$$

gives

$$
\begin{aligned}
&2\cos\mu_1 + \cos\mu_2 - \sqrt{3}\sin\mu_2 = 0, \\
&\cos\mu_0 + \sqrt{3}\sin\mu_0 + 2\cos\mu_1 = \frac{\lambda_p - \lambda_1}{3\alpha\beta^2}, \\
&\sin\mu_0 - \sqrt{3}\cos\mu_0 - 2\sin\mu_1 + \sin\mu_2 + \sqrt{3}\cos\mu_2 = 0, \\
&\cos\mu_0 + \sqrt{3}\sin\mu_0 - 2\cos\mu_1 + \cos\mu_2 - \sqrt{3}\sin\mu_2 = \frac{2\lambda_1}{3\alpha\beta^2}, \\
&\cos 3\theta_0 + \cos 3\theta_1 + \cos 3\theta_2 = 0, \\
&\sin 3\theta_0 + \sin 3\theta_1 + \sin 3\theta_2 = 0, \\
&\cos(3\theta_1) - \cos(3\theta_2) = 0, \\
&\cos(3\theta_1) - \cos(3\theta_0) = \frac{\lambda_1}{6\beta^3}.
\end{aligned}
$$

The last four equations are simple to solve and for $\lambda_P \neq 0$ give several solutions, all of which lead to equivalent $Q_i$, just with a few phase factors moved around. Choosing one of the solutions, we find $\lambda_1 = -9\beta^3$, $\theta_0 = 2\pi/3$, $\theta_1 = -2\pi/9$, $\theta_2 = 2\pi/9$.

The first equations have different solutions depending on the value of $\nu = 2\lambda_1/(\lambda_p - \lambda_1)$. Again, these solutions have some phase factors which can be shifted around. We pick one

particular set of solutions such that the solutions are continuous at finite $\nu$. If $\nu < 1/2$,

$$\alpha = \beta\left(\frac{2}{\nu} - 1\right),$$

$$\mu_0 = -\frac{2\pi}{3},$$

$$\mu_1 = \operatorname{atan}\left[\frac{1+\nu}{\nu-2}, \frac{\sqrt{3(1-2\nu)}}{\nu-2}\right],$$

$$\mu_2 = \operatorname{atan}\left[\frac{1+\nu-3\sqrt{1-2\nu}}{2(2-\nu)}, \sqrt{3}\frac{1+\nu+\sqrt{1-2\nu}}{2(2-\nu)}\right],$$

where $\operatorname{atan}[x, y]$ gives the $x$ and $y$ coordinates to allow the angle to be reconstructed without ambiguity. For $\nu > 1/2$

$$\alpha = -\beta\sqrt{\frac{4+\nu}{\nu}},$$

$$\mu_0 = \frac{\pi}{3} + \operatorname{atan}\left[\frac{\nu-2}{\sqrt{\nu(4+\nu)}}, -\frac{2\sqrt{2\nu-1}}{\sqrt{\nu(4+\nu)}}\right],$$

$$\mu_1 = \operatorname{atan}\left[-\frac{1+\nu}{\sqrt{\nu(4+\nu)}}, \frac{\sqrt{2\nu-1}}{\sqrt{\nu(4+\nu)}}\right],$$

$$\mu_2 = -\frac{\pi}{3} + \operatorname{atan}\left[\frac{1+\nu}{\sqrt{\nu(4+\nu)}}, \frac{\sqrt{2\nu-1}}{\sqrt{\nu(4+\nu)}}\right].$$

Taking $\beta \to 0, \alpha \to \infty$ but $\alpha\beta^2 \to$ const as $\nu \to 0$, we can keep the Hamiltonian well-defined here. The expressions for the two regions agree as $\nu \to 1/2$.

There are three special values $\nu = 0, 1/2, \infty$. Taking $\nu = 0$ yields $\pm H_P$ with the sign coming from that of $\alpha$. This behavior is analogous to the $\mathbb{Z}_2$ case, where the Ising point was recovered by taking one of the terms in $Q$ to zero. Taking $\nu \to \infty$ corresponds to $\lambda_P = \lambda_1$, the point where the $\tau_j + \sigma_j^\dagger \sigma_{j+1} +$ h.c. term vanishes. More mysterious is $\nu = 1/2$, which corresponds to $\lambda_p = 5\lambda_1$ in the Potts phase.

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
