# Peer review of "Self-dual $S_3$-invariant quantum chains"

_SciPost Physics, doi:SciPost Phys. 9, 088 (2020)_

## Round 2 · Referee Report · Dirk Schuricht · 2020-10-8

Report

The authors study the phase diagram of S_3 invariant spin chains. They find several gapped and gapless phases, locate the phase boundaries and characterise the phases using both analytical and numerical techniques. Particularly interesting is the supersymmetric c=3/2 phase they establish in part of the phase diagram. The results are very interesting and certainly should be published.

However, before publication the authors should revise the presentation of their results. In general I find the discussions quite short, with lots of references to other works and few numerical results. What I am lacking is a more detailed explanation as to how the authors reach their conclusions, and maybe also some background informations to make the presentation easier to access. More specifically I would like to suggest to address the following points:
1. The phase diagram (Fig. 1) is hard to read. Maybe a better presentation can be found by adding lines as guide to the eye (see, eg, Fig. 1 in PRB 53, 3304 (1996) for an example). Also the numerical values of the locations of the phase transitions could be included to arrive at a figure containing all main results.
2. At the beginning of Sec. 2.1 the authors state “No self-dual operator obeying the symmetries of H(θ) exists in the Potts CFT, with the least irrelevant such operator having dimension 14/5” but somehow the 2nd part of the sentence seems to contradict the first.
3. Figs. 2 are way too small. Also the plotted ratio R_1^1 is not defined. In the corresponding discussion around (10), what does the statement “are the dimensions of operator creating the states labelled by a and b respectively” mean, ie, which states are precisely considered in the specific cases shown in Fig. 2 and what are the corresponding operators? Also, how does one arrive at the predicted values of 17/12 and 10/3 given the stated scaling dimensions?
4. How are the “ratios” plotted in Fig. 4 defined? Do I understand correctly that the state E_a, here labelled by the momentum k, is varied?
5. In Sec. 3 the discussion of the possible marginal perturbations is very short and mostly refers to App. A. When looking at Tab. 1 I see three operators with eigenvalues D,D’=+1 and scaling dimension 2. However, there are other operators with D,D’=+1 and smaller scaling dimensions, so why do they not appear as perturbations since, as far as I understand, they should be self-dual as well?
6. When discussing Eq. (12) the authors state “One of these operators couples the two theories.” but how do the authors reach this conclusion? Similarly they later state “the Fermi velocity vTCI in Figure 4 quite clearly is vanishing”, but to me the ratio of velocities in Fig. 4 stays finite everywhere. So how should I understand this?
7. Finally, in Sec. 3 the authors suspect the existence of an incommensurate region, but some numerical data supporting this would be helpful.
8. In the caption of Fig. 5 the authors state “The incommensurability is readily apparent in the oscillations“ but an argument why oscillations are related to incommensurability would be helpful (after all there are many systems with oscillations that have other origins).
9. The authors suspect an incommensurate phase between θ ≈ −0.70π and θ ≈ −0.73π, but again some supporting data would be nice.

  • validity: -
  • significance: -
  • originality: -
  • clarity: -
  • formatting: -
  • grammar: -

Author:  Paul Fendley  on 2020-11-24

(in reply to Report 1 by Dirk Schuricht on 2020-10-08)

We thank the referees for their detailed comments on the paper. We have implemented as many of their suggestions as we could, and so we hope the paper is now ready for publishing in SciPost.

To reply to the referee's comments in more detail:

ref 1:

  1. We have improved the clarity of the figure by drawing the radial lines, thanks for the suggestion. We would prefer to leave the numbers off, as the figure is already on the cluttered side, and the precise values are not particularly important (and in a few cases, difficult to determine very precisely).

  2. We omitted the key word “relevant”, sorry! Now fixed.

  3. Plots were enlarged and R_1^1 added to caption. All the dimensions of operators were already listed in the text, and we now have put the numerical ratio in the text as well as the caption.

  4. We have added a more thorough explanation of how we extracted the velocity in the text (see also point 2 in our reply to referee 2).

  5. As we noted, operators appear in the effective field theory only if they obey all the symmetries of the lattice Hamiltonian, as well as being self-dual. To emphasise this point further, we added a sentence “For example, the self-dual dimension-7/8 operators have non-zero momentum and so violate translation symmetry.“

  6. As we noted, one of the marginal operators has dimensions 3/5 in the tricritical Ising CFT and 7/5 in 3-state Potts. Thus adding it to the action couples the two theories. We have rewritten this paragraph.

  7. As we discussed at the end of section 3, obtaining quantitative data on an incommensurate phase is extremely difficult. The many low-lying levels make DMRG borderline useless (it doesn’t converge well), and in ED the location (and the number) of the crossings changes as the system size is increased, making anything other than observing the phase borderline impossible. We did try, however, but don’t really have any plots that shed any light on the physics, only on our inability to extract any quantitative information on the incommensurate region. We thus would prefer not to add any more plots, because we would rather focus on the (large amount of) physics we do understand well.

  8. Not sure we can do better than the explanation we gave. As we noted, it’s not just the oscillations that imply the incommensurability, the excited-state level crossings also are a pretty strong sign. We are not aware of any (non-integrable) commensurate systems that exhibit level crossings. Note also that White and Affleck got the same result long ago, as we indicated.

  9. See our answer to comment 7.

ref 2:

  1. We thank the referee for this reference, as we were not aware of it. It reminded us of another paper by Bauer et al (1208.0343) with similar behavior as well.  We however have two marginal perturbations , so think it unlikely that the Lorentz symmetry is restored, given the numerics. However, the lessons from those papers are useful, and we rewrote that paragraph, including adding ``Indeed, such a marginally irrelevant perturbation occurs both in a c = 3/2 field theory [42] and a lattice model [43], also resulting in a Lorentz-symmetry-breaking perturbation. There, however, the effect is to restore Lorentz symmetry at large distances, whereas here the effect would be to leave the two effective theories decoupled.''

  2. Sorry for omitting how we did this measurement. As we now explain in the text, we fit the finite-size energies to a form $E/L + B/L^2$, what we meant by $L\to\infty$.

  3. Sorry again for omitting how we generated this plot. As we now explain, it came from DMRG. Unfortunately, we were unable to extract any more quantitative information, and we added “As the oscillations are rather small in magnitude (note the y-axis values on the log plot) we were unable to determine the period precisely. “

  4. This is a really great idea that did not occur to us. Unfortunately, the numerically adept person in this collaboration has left academia and so is not able to do the analysis. But we hope someone does! This phase we believe is definitely worth further study.

---

## Round 2 · Referee Report · Eran Sela · 2020-10-18

Report

In this work the authors provide an extensive study of a 1D model motivated by self-duality in the 3-state quantum Potts model. In fact, following their earlier paper Arxiv:1908.02767 entitled 'The “not-A”, RSPT and Potts phases in an S3-invariant chain', they combine this Hamiltonian with a second Hamiltonian that has the same symmetry, and study a rich family of interesting phases and phase transitions. The paper is nicely written, with a broad introduction, and a wide exposition to CFT methods, algebraic methods, and comparison to numerical input. I recommend the paper for publication after the authors have considered the optional remarks/suggestions below.

1. The c=3/2 critical theory, which the authors suggest to decompose into two CFTs with different velocities, reminds me the paper M. Sitte, A. Rosch, J.S. Meyer, K.A. Matveev, M. Garst
Phys.Rev.Lett. 102 (2009) 176404. While that model has a different symmetry, it has an emergent Lorentz symmetry. Is it possible that at very low energies the two velocities flow to equal values?

2. As a follow up, in Fig. 4 the ratio of two velocities is plotted. At what value of $L$ this is taken? (what does it mean $L \to \infty$?) It could be useful to actually show the $L$ dependence.

3. From figure 5 the authors claim that one can see oscillations in the spin-spin correlation away from the point $\lambda_1=\lambda_P/3$. First of all, at what value of $L$ is this calculation done? Now, can the authors extract numerically an actual period from a FT analysis of the data? Or is it just noise?
If this statement is correct and the oscillations are real, it will be useful to show how the wave number behaves as function of $\lambda_1$.

4. This is an optional suggestion: The separation of the c=3/2 theory into 4/5+ 7/10 sectors could be further tested by looking at the symmetry resoled entanglement (Phys. Rev. Lett. 120, 200602 (2018)). One could check how each symmetry sector scales with $L$. Since only one of the two theories carries the S_3 charge, one could fit to Eq.15 in Phys. Rev. Lett. 120, 200602 (2018), and demonstrate scaling with the c=4/5 theory.

  • validity: top
  • significance: top
  • originality: top
  • clarity: top
  • formatting: perfect
  • grammar: perfect

Author:  Paul Fendley  on 2020-11-26

(in reply to Report 2 by Eran Sela on 2020-10-18)

ref 2:

  1. We thank the referee for this reference, as we were not aware of it. It reminded us of another paper by Bauer et al (1208.0343) with similar behavior as well. We however have two marginal perturbations , so think it unlikely that the Lorentz symmetry is restored, given the numerics. However, the lessons from those papers are useful, and we rewrote that paragraph, including adding "Indeed, such a marginally irrelevant perturbation occurs both in a c = 3/2 field theory [42] and a lattice model [43], also resulting in a Lorentz-symmetry-breaking perturbation. There, however, the effect is to restore Lorentz symmetry at large distances, whereas here the effect would be to leave the two effective theories decoupled."

  2. Sorry for omitting how we did this measurement. As we now explain in the text, we fit the finite-size energies to a form E/L+B/L2, what we meant by L→∞.

  3. Sorry again for omitting how we generated this plot. As we now explain, it came from DMRG. Unfortunately, we were unable to extract any more quantitative information, and we added “As the oscillations are rather small in magnitude (note the y-axis values on the log plot) we were unable to determine the period precisely. “

  4. This is a really great idea that did not occur to us. Unfortunately, the numerically adept person in this collaboration has left academia and so is not able to do the analysis. But we hope someone does! This phase we believe is definitely worth further study.

---

## Round 3 · Referee Report · Dirk Schuricht (Referee 1) · 2020-11-28

Report

The authors have revised the manuscript and thereby increased its readability considerably. As far as I see they have addressed all the comments by the referees (except for one) and made good improvements accordingly. Thus in principle I support publication, but would like to ask again my previous question: At the end of Sec. 3 the authors state “the Fermi velocity vTCI in Figure 4 quite clearly is vanishing”, but to me the ratio of velocities in Fig. 4 stays finite everywhere. So how should I understand this? Maybe the authors still want to comment on this.

  • validity: -
  • significance: -
  • originality: -
  • clarity: -
  • formatting: -
  • grammar: -

Author:  Paul Fendley  on 2020-11-28  [id 1065]

(in reply to Report 1 by Dirk Schuricht on 2020-11-28)

Sorry for being slightly imprecise. We mean that it is quite clearly heading toward zero as $\theta$ is increased, and if extrapolated vanishes at approximately the value ($\sim .87\pi$) where the incommensurate phase begins. Although for reasons indicated in our first reply, doing numerics near or in the incommensurate phase is too difficult for us, we think the vanishing is a pretty reasonable inference from the figure. It also fits in with our other studies of the incommensurate phase, so I hope we can be forgiven for our slight imprecision.

---

## Round 3 · Author Response

We thank the referees for their detailed comments on the paper. We have implemented as many of their suggestions as we could, and so we hope the paper is now ready for publishing in SciPost.

---

## Round 3 · List of Changes

We listed almost all of the changes in the reply to the referees' reports. In addition, we made a few more minor pedagogical improvements and corrected a few typos.

---

## Editorial Decision

published